# Toxic Metals in Wild Ungulates and Domestic Meat Animals Slaughtered for Food Purposes: A Systemic Review

**DOI:** 10.3390/foods10112853

**Published:** 2021-11-18

**Authors:** Davies Veli Nkosi, Johan Leon Bekker, Louwrens Christian Hoffman

**Affiliations:** 1Department of Environmental Health, Tshwane University of Technology, Pretoria 0001, South Africa; bekkerjl@tut.ac.za; 2Department of Animal Sciences, University of Stellenbosch, Matieland, Stellenbosch 7602, South Africa; louwrens.hoffman@uq.edu.au; 3Centre for Nutrition and Food Sciences, Queensland Alliance for Agriculture and Food Innovation (QAAFI), The University of Queensland, Gatton 4343, Australia

**Keywords:** toxic metal, arsenic, lead, cadmium, mercury, hunting, harvesting, wild meat

## Abstract

The presence of toxic metals in harvested game meat is a cause for concern for public health and meat safety in general. Authorities and food safety agencies continue to develop guidelines and limits of the maximum allowable levels of toxic metals in food products. However, the situation is different for game meat products in developing countries, where a number of shortcomings have been identified. This includes a lack of game meat animal slaughter regulations, specific species’ product limits that have not yet been established and the continued use of hunting or game meat animals’ harvesting plans that could introduce the same toxic metals of concern. This review was conducted from English literature published between 2011 and 2021; it highlights the possible health effects and the shortcomings in the implementation of game meat safety production strategies for toxic metals (Arsenic, Lead, Cadmium and Mercury) in game meat animal production. Lead (Pb) remains the most significant threat for toxic metals contamination in game meat animals and the slaughter processes. In most developing countries, including in South Africa, the monitoring and control of these heavy metals in the game meat value chain has not yet been implemented.

## 1. Introduction

Worldwide, the shortage of protein sources as well as the high cost of red meat has contributed to the search for a sustainable supply of new sources of protein [1,2]. Ciobanu and Munteanu [3] state that at the turn of the century, the overabundance of game meat animals led to the identification of game as a possible future source of animal protein. Although game meat is a good source of protein, by nature of its killing and harvesting, it can lead to possible safety hazards that need to be controlled before such meat can be released for consumption by consumers and other carnivore animals [4,5,6,7,8]. While the welfare of the animal during the killing is always considered, compliance to food safety regulations is important [9]. Worldwide, the occurrence and existence of toxic metals in meat and its control remains a concern for authorities. Some of these toxic metals are naturally occurring in the environment, while others are introduced by processes of animal farming, industrial activity, urbanisation and mining [10,11]. Researchers concluded that the presence of toxic metals in meat must be investigated and controlled to ensure the safety of the meat products [12,13,14,15]. Several studies identified Lead (Pb), Cadmium (Cd), Mercury (Hg) and Arsenic (As) as leading game meat contaminants in terms of toxic metals [3]. Some of these toxic metals, such as lead (Pb), occur naturally in the environment in rocks, soil and the hydrosphere [16], while lead from contaminated foliage could contribute considerably to bioaccumulation in the meat of animals grazing there [17]. These risks could be identified and controlled by following good agricultural practices, such as ensuring that animals are not exposed to toxic metals during farming, improved farm management systems and water testing, feed protection from contamination and the selection of uncontaminated areas to keep game animals [18,19,20]. However, the control of these risks by means of good agricultural practices becomes challenging when game/bush meat is sourced via nonregulated (illegal) methods from non-farming systems, such as conservancies or bush/forests. It is important to note that the distribution of arsenic (As) is generally by natural processes; however, anthropogenic pollution is generally caused by mining, smelting activities, glass manufacturing, use of pesticides and preservatives that have traces of arsenic [13,20].

### General Health Effects of Toxic Metals

The consensus is that once these toxic metals are in high concentrations in the body, more adverse health effects can be expected [8,11]. While different exposure routes exist, ingestion and inhalation of high concentrations remain the most detrimental form of exposure [21]. Toxic metals, once ingested, tend to affect the major organs of the body due to bioaccumulation [21,22,23]. The body’s ability to excrete toxic metals is generally slower than the intake, thus leading to excessive toxic accumulation of these metals in the body [11,24]. Kamunda and Mathuthu [20] state that some of the long-term health impacts of over-accumulation by these metals may include carcinogenic effects, central and peripheral nervous system damage and blood circulatory effects.

The risk is no different with game/wild/bush meat, where the presence of these toxic metals in the meat continues to be a concern for authorities [25,26]. Confirmed by Falandysz and Szymczyk–Kobrzyńska [10] and Irschik and Wanek [12], some of the toxic metals found in harvested game meat could have been introduced by different game meat killing, hunting or harvesting methods. Similarly, other researchers [14,15,16] have documented the accumulative presence of trace metal elements in meat animals and have highlighted the need to ensure proper screening, monitoring and testing of meat products to ensure their compliance. Taggart and Reglero [27] noted that globally, developed countries have responded to the possible threat of the over-consumption of toxic metals by regulating the concentrations of toxic metals permissible in different food types, including meat. These limits include the cumulative concentration in the final products [10,26,28]. However, the situation could be different in developing regions/areas where mining activities, anthropogenic activities and different hunting and harvesting methods of game meat animals are adopted, as there is a probability of toxic metal traces being found in higher concentrations in food/s of animal origin [20,29]. This is worsened by the fact that generally preferred hunting bullets are made of lead and zinc [30]. Researchers Taylor et al. [5], Doabi et al. [18] and Kamunda et al. [20] noted that toxic metals in bullets can contaminate carcasses, especially around the bullet entry and exit points on the carcasses, thus increasing the meat safety risks of game meat. This then leads to unavoidable contamination of game meat animals during killing processes, especially when bullets made from lead (Pb) are used [31]. In many developing countries, the selection of bullets is left to the hunter; thus, in most cases, lead-made bullets are generally used as they are cheaper [26,32]. The European Union member states have recognized the threat from lead-containing bullets when used in hunting and are in the process of drafting regulations around the use of such bullets; these regulations have been reviewed in 2020 [33].

Figure 1 illustrates the health effects likely to develop in different systems of a human body due to excessive exposure to toxic metals.

Researchers Fowler et al. [20], Prashanth et al. [21], Mudgal et al. [23] and Durkalec et al. [24] agreed that some of the health effects of toxic metals as a result of the consumption of contaminated game meat could be long lasting and even have a negative effect on the development of babies. Some of the listed effects included increased child mortalities, reduced IQ and reduced reading and learning capabilities. While these effects were generally linked with exposed parents, the risks of developing cancer remain real in exposed adults [13,25]. This review highlights the need for a clear assessment and monitoring of the four toxic metals of interest (Lead, Cadmium, Mercury and Arsenic) in the game meat industry/supply chain, thus addressing the current lack of knowledge and research linked to the provision of safe game meat hunting or harvesting plans.

## 2. Materials and Methods

Due to a paucity of data on toxic metal contamination of game meat conducted in Africa, it was important for the purpose of this review that articles across the globe were included. This review was comprehended from English scholarly literature published in Science Direct, Google scholar and PubMed between 2011 and 2021, to ensure only recent information was included in the review. However, information pre-2011 was included in the introduction and discussions of the paper. Figure 2 describes the search strategy (adopted from http://www.prisma-statement.org/, accessed on 12 October 2020) used to identify relevant articles to be included in this review. The key search terms used were “Game Meat OR Toxic metals OR Lead (Pb) OR Arsenic (As) OR Cadmium (Cd) OR Mercury (Hg) AND Africa OR Europe OR South America OR North America OR Asia OR Australia OR Antarctica”. To assist with guidelines and standards, grey material from the websites of the Codex Alimentarius Commission (FAO), (www.fao.org, accessed on 3 November 2021), The Food Safety (European Commission) European Union (https://ec.europa.eu/food/overview_en, accessed on 17 November 2020) and the U.S. Food and Drugs Administration (www.fda.gov, accessed on 17 November 2020) were also searched for the latest update regarding the control and monitoring of toxic metals in food, with a focus on red meat products.

Records without a specific reference to toxic metals in wild animals and game meat, studies in languages other than English and postgraduate theses were excluded from this study as they did not relay to the objective of the study. Tables were created reflecting the aim, global legal limits and the findings and recommendations made by previous researchers.

## 3. Results

Although there is lack of sufficient data that addresses game meat animal species, the guidelines and standards set for maximum limits of toxic metals permissible in red meat from different countries and regions are depicted in Table 1. From the searched guidelines, it was clear that most developing countries were using the Codex Alimentarius Commission (CAC) guidelines to control toxic metals presence in food products, including meat.

Table 2 provides a summary of toxic metal studies conducted on meat animal products between 2011 and 2021 and their recommendations, taking into consideration the concentrations of the four most significant toxic metals (Pb, Cd, As and Hg).

Table 2 shows that studies were widespread across the globe and that developed countries contributed to most of the studies. For the search period of 2011 and 2021, countries where studies were done globally included Argentina (1), Australia (2), Canada (3), China (1), a combined study for EU countries (1), Germany (1) Hungary (2), Iran (3), Kuwait (1), Mexico (2), Namibia (1), New Zealand (1), Nigeria (3), Norway (1), Pakistan (1), Poland (6), Russia (1), Serbia (1), Slovakia (1), Spain (2), Sweden (1), Turkey (2), United Kingdom (2) and the United States of America (2). Most of the studies conducted were on game meat animals and general meat products; the most investigated toxic metals were Lead, Cadmium, Mercury and Arsenic. The general consensus was that the occurrences of toxic metals in meat animals and their meat could be contributed to by numerous factors, such as environmental contamination, feeds, water and the selection of a game meat animal killing methods utilised for the different game species.

## 4. Discussion

The presence of toxic metals in high concentrations could be dangerous to the environment and, subsequently, to the meat animal and, later, to consumers [23]. As stated by Mudgal and Madaan [23], Kanstrup and Thomas [33] and El-Kady and Abdel-Wahhab [36], in Figure 1, with extensive exposure, a significant number of health effects have been recorded. These health effects were observed in people traditionally reliant on game meat hunting and traditional foods [34,44]. Developed and some developing regions have set maximum limits of toxic metals in red meat products (Table 1). However, products of game meat origin are not included in the lists. This is a shortfall as it is widely known that game meat animals are mostly raised in the field, and their slaughter processes, by their nature, could introduce some hazards [81,82]. To add on top of these risks, some of the processes of game meat harvesting could be directly linked with the introduction of toxic metals such as Lead, Arsenic, Cadmium and Mercury in meat animals [24,83,84]

The greatest risk comes from the use of bullets made from lead, as these bullets tend to fragment and disperse across the harvested game animals, a situation that causes a wide distribution of physical and toxic metals contaminates in carcasses [48,85]. Figure 3 depicts x-rays of the spread of pellets in the head (left image) and along the lower neck, brisket and shoulder (right image) of impala (*Aepyceros melampus*) that were culled from a helicopter using a shotgun during commercial harvesting of game meat in South Africa.

Table 1 confirms that lead (Pb) concentrations greater than 0.10 Mg/Kg in meat products are considered dangerous, and meat products with concentrations greater than this must not be passed as safe for human and animal consumption [11,35,39]. The possible banning of lead bullets for game meat hunting is the most possible form of lead from bullets control [18,70,71]. The more a bullet fragments, the greater is the risk of spreading contamination to areas that were not initially contaminated. Using radiographic material and imaging, researchers showed that the selection of a bullet and the control or enforcement of its usage could be an effective control of Pb contamination caused by bullets in meat carcasses [71]. Lead has long been the preferred metal for constructing bullets due to its ability to expand upon hitting the target, thereby creating the hydrostatic shockwave required to create instant death. The hunting fraternity are changing from lead bullets to alternatives that are made of either 100% copper or copper alloys (95% Cu and 5% Zn); these bullets have the same or higher hydrostatic shock by unfolding when striking the target, but not breaking into pieces as lead bullets do.

In some of the studies documented in Table 2 on birds, the Pb poisoning was not only attributed to Pb from bullets/pellets, but also from birds consuming the spent pellets that were found to be lying on the ground/in the water bodies that the birds habituated, and this could be attributed to bioaccumulation [5,72]. These birds of prey and other predator animals could be utilized to indicate the levels of toxic metals in the surrounding areas [48,64,65]. An interesting report has also been published where cheetah during a rehabilitation phase died from Pb poisoning caused by the consumption of Guinea fowl killed with a shotgun, thus highlighting the eminent risk of secondary contamination of birds of prey or other predators/scavengers [86].

Arsenic (As) is also found on bullets and can be transferred to meat during hunting [49,58]. Its limit is also regulated in various countries where 1.0 mg/kg is the maximum permissible level in a red meat product (Table 1) [35]. In general, the choice of hunting bullets is important in the prevention of arsenic contamination of game meat animals [77].

Cadmium (Cd) is introduced to the environment through natural and man-made processes [13,87]. Game meat animals could be contaminated from eating contaminated grass around their habitats and thus transferring Cd to consumers and other scavenging animals [26,88]. Different countries and food safety organisations have regulated the amount and concentration of Cd in food products and, thus, control the distribution of the contaminated or suspected product (Table 1).

Mercury is ubiquitous in the environment and can be introduced by anthropogenic activities such as mining, agriculture and industrial areas [26]. Once meat animals are exposed to a contaminated area during grazing and general game-farming practices, traces of mercury could be ingested by the meat animal [30,89]. In developed countries, the levels of Mercury (Hg) in excess of 0.01 mg/kg in food including meat products is considered dangerous for consumption (Table 1), while other countries have no mercury limits in their food products [13]. This is because Mercury (Hg) is toxic, even in trace elements (although the mercury species differ in their toxicity, it is more toxic when in its organic form, with methylHg being more toxic than EthylHg) [90], and its presence in food must be monitored at all times [91]. Studies suggest that the environmental monitoring and control of processes linked with the harvesting of game meat can bring about control in the concentration of mercury in game meat products [14,77].

Despite the findings made by researchers on the presence of toxic metals in meat animals, meat and meat products, the adoption of the recommendations by stakeholders and role-players has been slow. From Table 2 above, the recommendations made can be summarised as follows:“One Health” concepts: According to the World Health Organisation (WHO) (www.who.org, accessed on 10 June 2021), “One Health is an approach to designing and implementing programs, policies, legislation and research in which multiple sectors communicate and work together to achieve better public health outcomes” [78]. The adoption of these concepts in the production of meat provides a structured way of investigating meat safety risks [92,93].Stakeholder’s involvement: Studies by Shrivastava and Shrivastava [94] and Duc and Toribio [95] concluded that when different stakeholders are identified or identifiable, information generally flows efficiently within different categories. This information could be for training purposes, ideas sharing and early warning systems to identify changes in contamination levels in the environment [96].Good Agricultural Practices (GAP): With the adoption of meat safety strategies during meat animals farming, a better foundation of toxic identification exists [97]. The same is applicable to game farming and the production of meat animals ready for slaughter. Measures such as compliance of feed, water and medication used; detoxification of industrial effluents before being released to the environments and monitoring of toxic metals in meat animals must be put in place to facilitate the process of meat animals production [98].Enforcement: Law enforcement would entail environmental pollution control, hunting and slaughter/processing control and the banning of generally used products with high concentration of toxic metals (e.g., fuel, paint) but, specifically, lead bullets and pellets used for hunting. This will ensure that secondary contamination of meat animals by these products is minimised [99,100].Bio-indicators: Environmental monitoring of toxic metals by using animals and plants as bio-indicators is an effective method of environmental pollution monitoring [101]. The concentration of toxic metals of interest could be found on leaves, plants, feathers, fur and skin of other animals [102,103]. The presence of specific plants in an ecosystem could be used to indicate toxic metals contamination in the environment [104].Risk assessment: Risk assessment from farm to fork could be the best tool to be used for the identification, evaluation and control of food hazards in a food supply chain [99,105]. The situation is no different for the game meat production systems and food safety hazards such as toxic metals must be controlled at farms, killing, slaughter and dressing processes and preparation by consumers [106]. For example, Food Safety Management Systems, such as ISO 22000 [9], require that suppliers of raw materials (including live animals sent for slaughter) must provide information such as risks to an extent that it will allow the next entity in the supply chain to conduct a hazard analysis.Monitoring: Similar to enforcement, monitoring of the presence and concentration of these toxic metals must be done to facilitate the detection levels of toxic metals in the environment (soil, vegetation, water and meat products) [107,108]. It is also a requirement of ISO 22000, the Codex Alimentarius standards, European Food Safety Authority and the United States Food and Drug Administration [36,38,39] that toxic metals hazards must be monitored at all times during food production. Known levels of toxic metals will assist with controls and decisions that need to be made to ensure safe meat [53,109]. While these could be done by different governments, industries must also play a significant role in developing monitoring systems.Further research: Wild angulate meat animals must be further researched as a means of ensuring that meat from these animals meet the requirements to be certified safe for human consumption [104]. Strategies employable to reduce meat contamination from toxic metals must be developed; these may include the promotion of the “One Health” approach in meat production from farm to fork [110].

## 5. Conclusions

Across the globe, there is sufficient evidence that confirms the presence of toxic metals in game and game meat, whilst there are no regulations/guidelines limiting the levels of toxic metals in meat produced for human and animal consumption. This situation leaves food authorities, consumers and governments vulnerable, and the situation cannot be overlooked. While toxic metals limits are documented for red (livestock) meat, fish and poultry meat, game meat has lagged behind. It is therefore important that game meat regulations are equally developed; these regulations must include the limits of toxic metals, especially in regions where game/bush meat is readily consumed. However, in implementing control measures, there are numerous obstacles to overcome. These may include a lack of funds to develop environmental monitoring and implementation of toxic metal management systems at farm levels, so concerted efforts to reduce the occurrence and introduction of toxic metals in meat from farm to fork is needed. The research recommendations that will influence control strategies mainly provide for a One Health Approach, stakeholder’s involvement, good agricultural practices, enforcement, use of bio-indicators, risk assessment, monitoring, awareness programs and further research. It is clear that more research is needed globally, but especially in regions which rely on game meat as a major source of protein.

## Figures and Tables

**Figure 1 foods-10-02853-f001:**
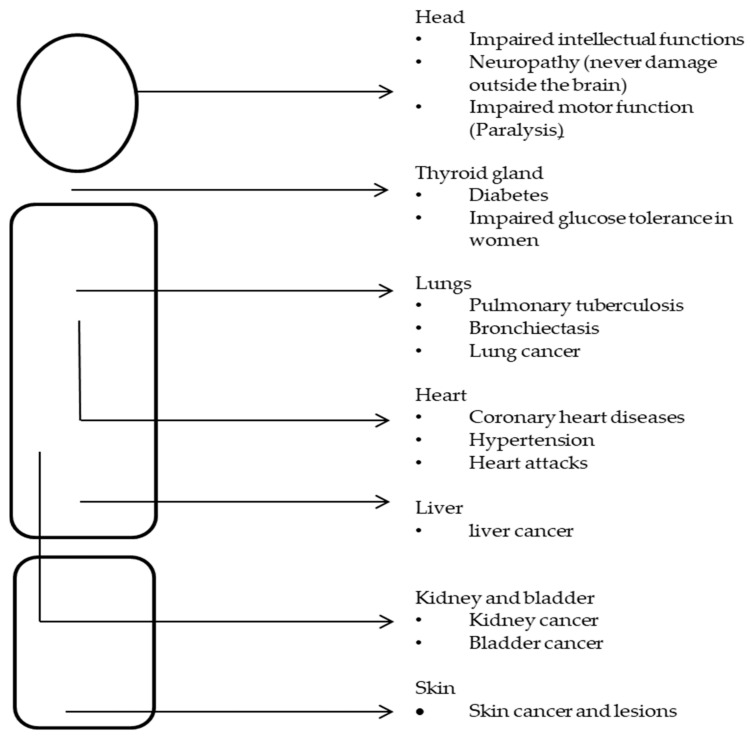
Human health effects of exposure to toxic metals. Figure developed from information sourced in [21,22,23,34].

**Figure 2 foods-10-02853-f002:**
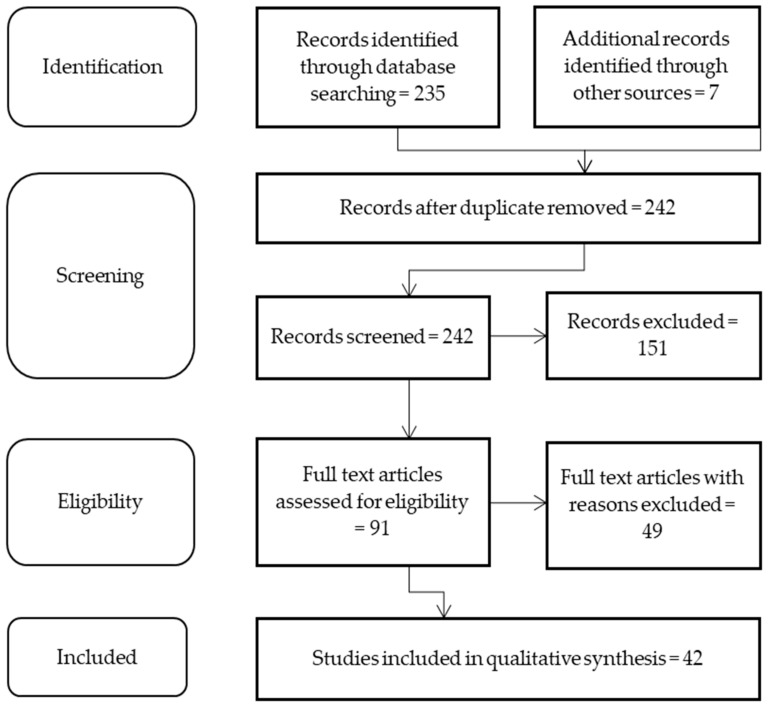
Methodology of search followed during the review process (available from: http://www.prisma-statement.org, accessed on 12 October 2020).

**Figure 3 foods-10-02853-f003:**
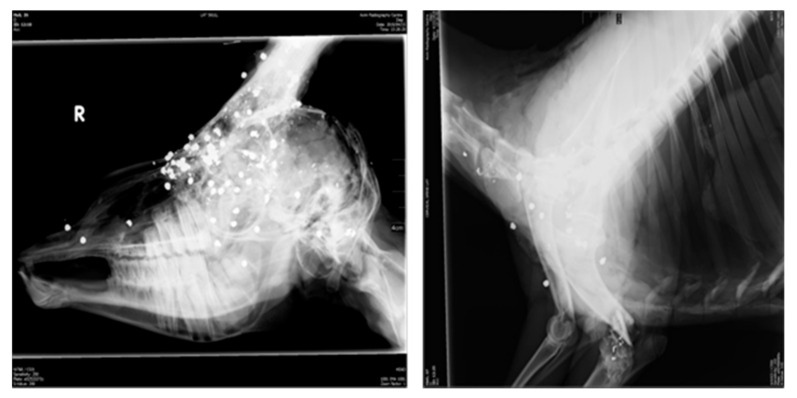
Impala head (**left**) and impala neck (**right**) showing the spread of shotgun lead bullets fired from a helicopter during a commercial harvest. **R** = Picture taken from the right side when facing the animal.

**Table 1 foods-10-02853-t001:** Maximum limits for toxic metals in red meat products (presented in mg/kg).

Country	Type	Arsenic (Semi-Metal)	Lead	Mercury	Cadmium	Source(s)
South Africa	Regulation	1.0	0.10	-	0.050	[35]
EU countries	Regulation	0.3	0.10	1.0	0.050	[36]
Republic of China	Regulation	0.5	0.5	0.05	0.1	[37]
Australia and New Zealand	Regulation	0.01	0.10	0.002	0.05	[11]
United States of America	Regulation	0.01	0.01	0.01	0.01	[38]
Countries without specified limits	Codex Alimentarius standards	2.1	0.10	-	0.05	[39]

Notes: No literature could be found for any of the other African countries; countries with no regulation generally used the codex guidelines.

**Table 2 foods-10-02853-t002:** Summary of studies conducted in toxic metals in red meat including game meat between 2011 and 2021.

Country	Aim	Species	Study Findings and Recommendations	Sources
Argentina	To investigate the presence and concentrations of toxic metals in waterfowl (*Anseriformes*) hunted.	Waterfowl (*Anseriformes*)	Pb pollution from hunting ammunition is increasingly recognized as a significant local problem, impacting wildlife. Regulating the use of Pb free bullets must be compulsory. Awareness on the risks of using these hunting methods must be advocated.	[40]
Australia	To analyse trace elements of Cd, Pb, Hg and As in the muscle, liver and kidney of sheep commercially farmed in Australia.	Sheep (*Ovine aries*)	Concentrations of the elements considered were within current acceptable ranges. This could be caused by the role and influence of environmental control employed by sheep farmers.	[41]
Australia	To investigate the levels of Pb on carcasses shot as part of a culling program for wild animals’ control in Australia.	Game animals	Bullet fragments may present a larger reservoir of toxic material in Australia than in most countries in the world. Bullets are likely to be one of the greatest sources of Pb that is knowingly discharged into the environment in Australia. One Health in the game meat industry must be followed in Australia. Research is needed to investigate Pb exposure in Australian people and wildlife arising from the use of Pb-based bullets.	[42]
Canada	To investigate trace elements of As, Cd, HG and Pb concentrations in the breast muscle of harvested auks from Newfoundland.	Birds (*Uria* spp. and *Alca torda*)	27% of auks had trace element (As, Pb and Cd) concentrations above Canadian food safety standards. Investment in environmental monitoring plans is recommended. Government-funded and -driven monitoring plans must be integrated into the meat safety program.	[43]
Canada	To measure the concentration and bio-accessibility of toxic metals in traditionally consumed foods	Wild game-Moose (*Alcer alces*), Deer (*Cervidae*) and Rabbit (*Oryctolagus cuniculus*)	As, Cd, Hg, Se, Mn and Cu are generally found in in wild game meant. As, Cd and Hg concentrations were observed in the kidneys and livers of wild game meat, though in low concentrations.	[44]
Canada	To investigate the levels of Pb poisoning in Moose resulting from the consumption of ammunition residues in the tissues of big game killed by hunters.	Moose (*Alces alces*)	Pb-made ammunition is widely used in big game killing and is the most important source of contamination for both wildlife and human health. Ammunition used during hunting must be regulated to protect scavengers, predators and human exposure to Pb.	[45]
China	To investigate the role played by environmental pollution and its contamination of game animals	Wild game	Areas with high mining and smelting activities contributed the most to contamination of toxic metals in wild animals. Areas with high traffic or transportation volumes contributed significant concentrations of toxic metals, such as Pb. Traces of toxic metals on dead carcasses and bones of dead animals could be an indicator of environmental pollution and long-term exposure.	[46]
EU Countries	To investigate the possible setting of maximum levels for lead in game meat in EC regulations.	Game meat animals species	Maximum limits on game meat animals must be regulated, similar to domesticated animals. Pb ingestion by scavengers and human beings would be reduced by hunters’ use of non-lead ammunition. There is a great need to replace lead ammunition with non-toxic alternatives	[47]
Germany	To investigate the differences in the fragmentation patterns of Lead and Lead-free bullets in real hunting conditions	Game meat animals hunted.	Lead-based bullets cause a broad contamination of the carcass and the viscera with bullet material. Lead-based bullets fragmented more than lead free bullets upon hitting an animal and thus spread the contamination The replacement of lead-based with non-lead-based bullets must be implemented for all meat animals killing.	[48]
Hungary	To evaluate the As, Cd, Pb and Hg concentration in different predator birds: Hen harrier and Marsh harrier around Hungary.	Hen harrier (*Circus cyaneus*) and Marsh harrier (*C. aeruginosu*)	Predator birds provided a good indicator of environmental contamination by As, Cd, Pb and Hg. Birds feather testing is a useful method for monitoring the presence of toxic metals, which can be found in the air and could be transported atmospherically. Predator birds can be poisoned by consuming other birds that are contaminated. The enforcement and ban of Pb-made ammunition usage is the best way of ensuring that the environment is less contaminated.	[49]
Hungary	To investigate the concentration of Cd, Pb, Hg and As in the muscle tissue of Roe deer.	Roe deer (*Capreolus capreolus*)	The average Pb concentration (0.48 ± 0.21 mg/kg wet weight) exceeded the regulated maximum limit of 1.0 mg/kg. The measured concentration of As (0.27 ± 0.20 mg/kg wet weight) in the roe deer meat might not pose any health risk for the human consumers. Monitoring of toxic metals in game meat and the restrictions of contamination in meat is important to ensure safe meat production.	[50]
Iran	To estimate the health risks of exposure to Cd, Cr, Fe, Pb, and Zn due to the consumption of pectoral muscle of mallard and pochard.	Mallard (*Anas platyrhynchos*) and pochard (*Aythya ferina*)	There is a potential human health risk due to the presence of toxic metals in the flesh of mallard and pochard in the Southeastern Caspian Sea region of Iran. Toxic metal levels in pochard flesh were not unsafe for consumption, as levels were below 1 mg/kg. Monitoring must ensure that this daily exposure is not exceeded.	[51]
Iran	To determine the concentration of Cr, Pb, and Cd in chicken tissue from Mashhad, North-East of Iran.	Chicken (*Gallus gallus domesticus*)	The presence of some toxic metals in chicken highlights its public health risk in this region of Iran. Effective monitoring procedures and surveillance programs should be applied by the authorities of the regional veterinary organization.	[52]
Iran	To review the level of food contamination with toxic metals in Iran.	Chicken (*Gallus gallus domesticus*)	Environmental contamination Pb and Cd concentrations were higher in samples of poultry (higher than 1.0 mg/kg). More effective monitoring procedures and surveillance programs should be applied by the authorities of regional veterinary organizations on meat animals before slaughter.	[53]
Kuwait	To investigate the presence of Hg, As, Pb, Cd, and Cr in slaughtered sheep (*Ovine aries*) at abattoirs in Kuwait.	Sheep (*Ovine aries*)	The concentrations of all toxic metals, except Cr, exceeded the maximum permissible limits set by various international food agencies. Monitoring systems must be compulsory for all sheep meat. Environmental monitoring to identify contaminated sites must be done at each sheep farm.	[54]
Mexico	To investigate the levels of Pb in Blue-Winged teals.	Birds (Teals) (*Anas discors*).	Pb concentrations in Blue-Winged teals exceeded 5.0 mg/kg dry weight in 34% of tested tissues. The consumption of Blue-Winged teals could lead to exposure to Pb by scavenging animals as well as human beings. Enforcement against the use of Pb bullets must be done across areas selected for hunting purposes.	[55]
Mexico	To investigate the concentrations of As, Cd, Cr, Hg, Pb, and Se in pectoral muscle of Woodcock.	Woodcock (*Scolopax*)	Hg and Cr concentrations were below current biologically significant thresholds. The levels of Pb and Cd were significantly higher in muscles of woodcock as compared to the threshold of 10 μg/g. Environmental monitoring of toxic metals must be conducted to identify areas that are contaminated and thus could contribute to wild meat animals’ contamination.	[56]
Namibia	To assess toxic metal (Pb and Cd) values in meat and offal from harvested springbok.	Springbok (*Antidorcas marsupialis*)	Springbok had measurable (above the detection limit) levels of Pb and Cd: 0.1 mg/kg and 0.05 mg/kg, respectively. Extensive risk assessments for Pb and Cd are needed in the Namibian wild environment. Health and stakeholder awareness programs should be developed for regular monitoring of Pb and Cd in water, soil and plants on farms to assess the impact and trends of toxic metals in human and animal health.	[57]
New Zealand	To investigate the concentration of 22 elements (including Fe, Ca and Se) and selected organochlorines in Mutton birds over two years.	Mutton bird (*Puffinus griseus*)	The concentration of Fe, Ca and Se in Mutton bird was higher than that in domestic land animal meats reported in literature. Mutton bird meat is high in essential nutrient elements and of a low toxicological risk. A monitoring program for contaminants in Mutton birds was recommended.	[58]
Nigeria	To evaluate and compare the levels of toxic metal contamination in the muscle, liver and kidney of both fresh and smoke-dried porcupine meats sold in Edo State, Nigeria.	Porcupine (*Hystricomorph Hystricidae*)	The levels of Pb in carcasses were linked to the bullets used. Domestication of porcupine by public and private sectors is a viable and environmentally-friendly alternative to prevent contamination, as compared to uncontrolled farming environments.	[59]
Nigeria	To assess concentration levels of Cd, Pb, Cu and Mg in cattle slaughtered in Jos North and South of Plateau State, Northern Nigeria.	Cattle (*Bos taurus*)	High levels of Pb in all the blood samples of cattle screened in both Jos North and South areas are due to high mining activities. Monitoring of Pb in animals before slaughter must be done to ensure compliance to only slaughter animals with low levels of Pb. Regulations must be developed to promote environmental monitoring at farms.	[60]
Nigeria	To determine the concentrations of Cd, Fe, Mn, Pb and Zn in the kidney and liver of slaughtered cattle.	Cattle (*Bos taurus*)	Toxic metal concentrations in kidney and liver samples were within the maximum permissible limit of the European Commission (EC) and FAO/WHO standards. Monitoring should not be only on the liver and kidney but on all offal meats sold across all abattoirs within the country to ensure that only safe meat is sold for human and animal consumption.	[61]
Norway	To study the concentration of toxic elements in semi-domesticated reindeer in Norway.	Reindeer (*Rangifer tarandus*)	Cd and As were the only toxic elements positively identified in the liver and meat of Reindeer. Reindeer meat from Norway can be consumed without any risks of human exposure to toxic metals. Maintenance of current toxic metals monitoring plans in domestic reindeer remains important to prevent contamination.	[62]
Pakistan	To detect the concentrations of various toxic metals in meat, liver, milk and fodder of buffaloes.	Buffalo (*Bubalus bubalis*)	Levels of Cu and Cd were higher in samples as compared to values reported in the literature. Detoxication of industrial effluents being used for irrigation purposes is highly recommended. Industrial effluents result in the transfer of metals to meat animals’ food chain.	[63]
Poland	To determine total Hg concentrations in the liver, kidney, semimembranosus muscle and brain of raccoons originating from the Warta Mouth National Park (WMNP) in north western Poland.	Raccoons (*Procyon lotor*)	Omnivore animals, such as raccoons, can be used as good bio-indicators for environmental contamination. Hg levels in the liver and kidneys of raccoons ranged between 0.03–18.45 mg/kg dry weight (dw). with a mean value of 2.99 mg/kg (dw). Raccoons from high mining areas contributed to the higher concentrations of Hg. Environmental controls and monitoring must be conducted to ensure proper and effective implementation plans to prevent animal and environmental contamination.	[64]
Poland	The estimation of Hg, Pb and Cd concentrations and the determination of relationships between these elements in the brains of mesocarnivores (Wild badger and fox), road-killed or hunted animals.	Wild badger (*T. taxus*) and Fox (*Vulpes vulpes*)	Pb-based ammunition is a significant source of toxic metals in scavengers road killed or hunted. A significant correlation exists between Pb and Cd levels in the fox brain. The background levels for brain Pb and Cd in meso-carnivores were between <0.50 and <0.04 mg/kg dry weight (dw), respectively. The control of environmental pollution is an important step of contamination control.	[65]
Poland	To determine the concentrations of total Hg in samples of liver, kidney and skeletal muscle of red foxes.	Red foxes (*Vulpes vulpes*)	Red fox exhibits a measurable response to Hg environmental pollution. Red fox can be used as a bioindicator of Hg environmental contamination in Terrestrial Ecosystems of North-Western Poland.	[66]
Poland	To assess the accumulation of Cd, Pb and Hg in the tissues of roe deer, red deer and wild boar from selected major industrial areas.	Roe deer (*Capreolus capreolus*), Red deer (*Cervus elaphus*) and Wild boar (*Sus scrofa*)	The highest concentration of toxic elements was found in roe deer and wild boar from the Upper Silesia Region, which indicates a high contamination of the environment compared to other areas. The concentration of toxic elements in the tissues of game animals may be a useful source of information on the quality of ecosystems in which they live.	[30]
Poland	To investigate Pb in game bird meat as a risk to public health.	Game bird meat	Shotgun pellets are the main source of Pb contamination of game animal tissues. Since each game bird shot with Pb pellets should be handled as potentially dangerous to humans, it should be excluded from the human diet. The use of Pb pellets in bird hunting should be banned.	[67]
Poland	To investigate the concentration levels of Pb, Cd and Hg in livers and kidneys of red deer harvested around industrial smelting plants.	Red deer (*Cervus elaphus*)	Smelter/environmental pollution contributed to high levels of Pb and Cd in red deer carcasses. Bullets type also plays a role in contaminating red deer. The environmental effect on game meat animals must be investigated with any game meat harvesting plan.	[68]
Russia	To investigate the levels of dioxins, dl-PCBs, Cd and Hg in reindeer meat, liver and kidneys samples.	Reindeer (*Rangifer tarandus*)	Consumption of reindeer meat from the studied regions does not bare a significant risk for health. Long-term eating of a reindeer’s liver and kidneys could be risky because of the presence of Pb that were exceeding permissible Codex limits three-fold. To reduce environmental contamination, the control of hunting by banning the use of Pb bullets is essential.	[69]
Serbian	To evaluate the concentration of environmental contaminants Pb, As, Hg, Cd and Cu in tissues of free-living roe deer in Serbia.	Roe deer (*Capreolus capreolus*)	High concentrations of Pb in two muscle samples are most likely due to the proximity to the killing wound area. The presence of some elements in the tissues of roe deer suggests the necessity of further research aimed at identifying the source of contamination in order to preserve the health of both humans and animals. A control program should involve all stakeholders in the wildlife management, such as veterinary officials, as well as hunters and other subjects involved in the game meat chain.	[70]
Slovakia	To present the concentrations of Cd, Co, Cu, Hg, Pb and Zn in the kidneys, liver and muscles of wild boar	Wild boar (*Sus scrofa*)	Levels of toxic metals in kidney, liver and muscle tissues may influence the fitness of such meat being certified safe for human consumption. Cd and Pb reached concentrations that exceed the permissible accumulation in meat (thresholds are only established for farm-reared meat) and increase the risk of the consumers.	[71]
Spain	To study the interactions between glutathione, superoxide dismutase and peroxidase antioxidants and to evaluate their role in fighting Pb-induced oxidative stress in wild ungulates.	Red deer (*Cervus elaphus*) and Wild boar (*Sus scrofa*)	Red deer and wild boar exposed to Pb may have a low or impacted natural antioxidants production in the animal. Meat animals’ exposure to Pb (natural or cumulative) must be prevented during meat animal tending, as they contribute to natural antioxidation processes in live animals.	[72]
Spain	To analyse the presence of Pb and its relationship with Pb-based ammunition in terrestrial game birds: Woodpigeons and Rock doves.	Woodpigeons (*Columba palumbus*) and rock doves (*Columba livia*)	Common woodpigeons and rock doves from Madrid were found to have high concentrations of Pb in their livers as a result of gun pellets consumption. Both birds species can be considered to be good bioindicators of Pb contamination in rural and urban environments. The use of Pb-made pellets for hunting purposes should be stopped.	[73]
Sweden	To investigate game meat hazards in moose killing, handling and processing in Sweden.	Moose (*Alces alces*)	Game hygiene handling can bring about the reduction in meat hazards. Pb remains the most significant threat to game meat animal’s safety in Sweden. The majority of meat derived from game meat is shared by hunters, farmers and their families. Stakeholders’ engagement and education on proper bullets selection especially (Pb-free bullets) must be done to facilitate the production of safer wild meat products.	[74]
Turkey	To evaluate the concentrations of As, Cd, Hg and Pb in the kidney, liver, lung, muscle and brain of slaughtered cattle from Sivas, Turkey.	Cattle (*Bos taurus*)	As, Cd, Hg and Pb were found to be high in cattle organs’ liver and kidney. Consumers have no knowledge of the quantity of the toxic metal in the content of these products and the public health risk they carry. Poor animal husbandry could lead to an increased concentration of toxic metals in cattle. This included feeding cattle contaminated feed, poor environmental monitoring for toxic metals and farming practices on previous mining or industrial sites that could be contaminated.	[75]
Turkey	To investigate concentrations of Mn, Cu, Zn, Cd, Hg, Pb and Se in Wild boars and brown hares.	Wild boars (*Sus scrofa*) and brown hares (*Lepus europaeus*)	Areas around the shot site had high concentrations of toxic metals. Determining the concentration of toxic metals in meat animal before slaughter should be undertaken as a prerequisite before slaughter.	[76]
United Kingdom	To investigate the effects of lead from ammunition on Wildfowl (*Anatidae*) birds.	Wildfowl (*Anatidae*)	Pb poisoning of birds is likely to occur wherever Pb ammunition is used and a pathway of exposure exists. Cases of Pb poisoning in new species and countries suggest a growing contamination trend in the United Kingdom. Pb-based ammunition usage must be banned in the United Kingdom.	[77]
United Kingdom	To estimate potential risks to human health in the UK from dietary exposure to Pb from wild game birds killed by shooting.	Game birds	The lead contamination may counter the benefit of game meat protein. Ammunition-derived lead is now a significant cause of dietary lead exposure in groups of people who eat wild game meat frequently. Pb monitoring as a compulsory environmental management plan is recommended. The compulsory replacement of all Pb-made bullets used for meat animals killing must be approved.	[78]
United States of America	To determine the levels of Hg, Se and As in muscle and liver tissues of game meat animals.	Wild pigs (*Sus scrofa*), Waterfowl (*Anseriformes*) and grey squirrels (*Sciurus carolinensis*)	The concentrations of six trace elements (Cr, Cu, Zn, Se, Hg, Cd) were above detection levels in products. Wide-scale sampling for contaminants in game meat animals is needed to better understand routes of contaminant movement and potential areas (or species).	[79]
United States of America	This review discusses Pb exposure routes, effects of Pb toxicity and the distribution of Pb in American woodcock.	Woodcock (*Scolopax minor*)	Environmental monitoring of Pb should be done at areas where game meat animals/birds are kept. More investigation must be done to investigate the relative absorption of Pb by woodcock birds.	[80]

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
