# Peer review of "Toxic Metals in Wild Ungulates and Domestic Meat Animals Slaughtered for Food Purposes: A Systemic Review"

_foods, 2021, doi:10.3390/foods10112853_

Round 1

Reviewer 1 Report

This manuscript is well-organized. The articles regarding hazardous heavy metals are appropriately collected, and the essential contents are discussed.

A minor revision of a few mistakes as indicated below:

line 209: methylHg

line 356, 360: Is there any other department/authority as the authors instead of name of countries?

line 358: This writing of reference is wrong. Need to be corrected.

Author Response

Response to Reviewer 1 Comments

Point 1: This manuscript is well-organized. The articles regarding hazardous heavy metals are appropriately collected, and the essential contents are discussed.

Response 1: Thank you, for the positive comments, we really appreciate them.

Point 2: line 209: methylHg

Response 2: Noted, the spelling of “Methylmercury” has been fixed.

Point 3: line 356, 360: Is there any other department/authority as the authors instead of name of countries?

Response 3: Noted, Names of issuing departments in respective countries has been included in the reference list.

Point 4: line 358: This writing of reference is wrong. Need to be corrected.

Response 4: Thank you, this reference (and the entire list) has been corrected. All species names written in italic, year bolded and references not in the text have been removed.

Reviewer 2 Report

Summary:

This review is a valuable addition to the literature on the safety of game (and domestic) meat. The review reports on scientific papers that study the presence (or at least measuring) heavy metals in wild game (and sometimes domesticated) meat. Overall the review appears to be systematically carried out; however reported findings in Table 2 sometimes appear subjectively selected or as secondary information within a paper. In addition, the background information is sometimes taken secondarily from sources. Please find and report primary sources and their results only within the literature review as well as the background section. One other slight drawback is the very limited South African perspective. I applaud the authors on conducting such a review, as pointed out often game meat is overlooked. Because of this fact, this research is applicable worldwide, particularly in regions reliant on game meat as a major source of protein. For this fact, I request that the authors broaden how they frame their research. Please see below for more specific comments.

Comments:

The manuscript has numerous minor typos and grammatical errors. Please refer to the annotated text for some of the necessary corrections.

The review is broad and global, yet the manuscript is very South African focused. I recommend reframing the paper in a broader sense, so as to reach the wider audience for which this paper is also applicable.

Please move the “general health effects” section into the Introduction and please break up the Introduction text into paragraphs, where appropriate.

Citation #27 is not appropriate for the “general health effects” section, as it is a secondary source for the information in Figure 1. Please find empirical research studies or other relevant sources (from health authorities) for information pertaining to human health. Please use studies on human nutrition/human health when citing possible negative effects of heavy metals on human health.

M&M section should be section 2 (not 3)

Please provide a justification for only including literature within the last 10 years. If there is so little information on this topic available, perhaps older sources could be of value as well. Although there could be some legitimate reasons to limit your search. Please explain/justify in the M&M section.

Table 1 - please comprehensively list all countries identified in the review, or at least state that all others not listed employ CAS, if this is the case. I believe some countries with regulations are missing. The table in its current form appears to be coincidental in which countries/regions it lists.

Table 2 - study findings can often be shortened. Only primary findings (and recommendations based on those primary findings) should be reported. Secondary findings (those cited within a piece of literature) and other broad information should be left out. The findings appear to be missing from a couple of the pieces of literature. As it currently stands the table appears to be subjectively constructed by the authors as sometimes reported findings do not match with the stated research objective of the paper. Please revise and be more systematic in retelling results.

Please expand on how to reduce environmental exposure to game animals. Not all animals are farmed. There is literature out there on reducing exposure to harvesters/consumers who rely on unfarmed game meat.

Please start a new paragraph with “The greatest risk comes from using lead bullets…”

Please include a source for Figure 3. If the source is the authors, please provide a little bit more context around how, when and why the image came to be taken.

Please include a summary of the review findings in the abstract; I recommend a sentence on how contamination occurs and what recommended preventitive actions are.

Possible (applicable) literature:

Juric et al. (2018) “Risk assessment of dietary lead exposure among First Nations people living on-reserve in Ontario, Canada using a total diet study and a probabilistic approach”

Juric et al. (2017) “A total diet study and probabilistic assessment risk assessment of dietary mercury exposure among First Nations living on-reserve in Ontario, Canada”

Laird and Chan (2013) “Bioaccessibility of metals in fish, shellfish, wild game, and seaweed harvested in British Columbia, Canada”

Author Response

Response to Reviewer 2

General comments:

This review is a valuable addition to the literature on the safety of game (and domestic) meat. The review reports on scientific papers that study the presence (or at least measuring) heavy metals in wild game (and sometimes domesticated) meat. Overall the review appears to be systematically carried out; however reported findings in Table 2 sometimes appear subjectively selected or as secondary information within a paper. In addition, the background information is sometimes taken secondarily from sources. Please find and report primary sources and their results only within the literature review as well as the background section. One other slight drawback is the very limited South African perspective. I applaud the authors on conducting such a review, as pointed out often game meat is overlooked. Because of this fact, this research is applicable worldwide, particularly in regions reliant on game meat as a major source of protein. For this fact, I request that the authors broaden how they frame their research. Please see below for more specific comments.

Response to general comments: Thank you, for the positive comments.

Comments:

Comments 1: The manuscript has numerous minor typos and grammatical errors. Please refer to the annotated text for some of the necessary corrections.

The review is broad and global, yet the manuscript is very South African focused. I recommend reframing the paper in a broader sense, so as to reach the wider audience for which this paper is also applicable.

Response 1: Thank you, indeed most of references have been made to South Africa; We have reworked the manuscript to highlight the global relevance of the review. All references to regional practices have been removed unless it was in a discussion of the findings.

Comment: Please move the “general health effects” section into the Introduction and please break up the Introduction text into paragraphs, where appropriate.

Response: The introduction has been broken down to two sections. The second section includes the general health effects, presented in figure 1.

Comment: Citation #27 is not appropriate for the “general health effects” section, as it is a secondary source for the information in Figure 1. Please find empirical research studies or other relevant sources (from health authorities) for information pertaining to human health. Please use studies on human nutrition/human health when citing possible negative effects of heavy metals on human health.

Experimental health effects attributed by toxic metals have been considered and used to describe the likely health impacts from prolonged exposures to heavy metals. Citation 27 has been removed in the description of heavy metals health effects.

M&M section should be section 2 (not 3)

Response Thank you, M&M section correctly labelled 2.

Comment: Please provide a justification for only including literature within the last 10 years. If there is so little information on this topic available, perhaps older sources could be of value as well. Although there could be some legitimate reasons to limit your search. Please explain/justify in the M&M section.

Response: Justification for the period has been added under M&M. Due to paucity of information on toxic metal in game meat animals, it was picked up that information was being repeated and it would be beneficial to the manuscript to include only recent information on the subject.

Table 1 - please comprehensively list all countries identified in the review, or at least state that all others not listed employ CAS, if this is the case. I believe some countries with regulations are missing. The table in its current form appears to be coincidental in which countries/regions it lists.

Response: Thank you, a statement has been included that countries generally not listed were using CAC guidelines. Regulations of major or developed countries were included in table 1.

Table 2 - study findings can often be shortened. Only primary findings (and recommendations based on those primary findings) should be reported. Secondary findings (those cited within a piece of literature) and other broad information should be left out. The findings appear to be missing from a couple of the pieces of literature. As it currently stands the table appears to be subjectively constructed by the authors as sometimes reported findings do not match with the stated research objective of the paper. Please revise and be more systematic in retelling results.

Response: Thank you, where appropriate we have revisited our references as pertaining to the information they contain and their relevance to the objective of the Review; those that do not meet the criteria have been removed.

Please expand on how to reduce environmental exposure to game animals. Not all animals are farmed. There is literature out there on reducing exposure to harvesters/consumers who rely on unfarmed game meat.

Response: Thank you the table has been reworked and improved. Where appropriate we have included references to harvesters/consumers of non-farmed/free roaming animals

Please start a new paragraph with “The greatest risk comes from using lead bullets…”

Response: This section has been included into a new paragraph.

Please include a source for Figure 3. If the source is the authors, please provide a little bit more context around how, when and why the image came to be taken.

Indeed, more information about Figure 3 has been included in the discussion of the manuscript. Figure 3 was created by the authors during a field examination. This study is part of a larger study on the slaughter of Impala (Aepyceros melampus) for commercial purposes.

Please include a summary of the review findings in the abstract; I recommend a sentence on how contamination occurs and what recommended preventative actions are.

Response: Thank you, a summary of the review results has been also included in the abstract

Possible (applicable) literature:

Juric et al. (2018) “Risk assessment of dietary lead exposure among First Nations people living on-reserve in Ontario, Canada using a total diet study and a probabilistic approach”

Juric et al. (2017) “A total diet study and probabilistic assessment risk assessment of dietary mercury exposure among First Nations living on-reserve in Ontario, Canada”

Laird and Chan (2013) “Bioaccessibility of metals in fish, shellfish, wild game, and seaweed harvested in British Columbia, Canada”

Response: Thank you, the suggested articles have been read and included in the manuscript. Very interesting articles, we really appreciate.

Reviewer 3 Report

The review gives interesting outcomes on the global bibliography referred to the presence of toxic metals in game meat, encouraging the need to establish regulations relating to the presence of these contaminants. The novelty of the paper relies on the first systemic review on the incidence of toxic metals in this type of products in South Africa.

However, the present work presents several weaknesses regarding the complementarity of the scope of the study with what was reported in the main document. Furthermore, several improvements regarding the method of investigation used and the discussion of the results are needed. All my suggestions are listed below:

Introduction

The overall introduction is poor of references and gives unnecessary information, considering the aim of the work. 

Lines 65-70: Considering the title of the MS, it could be a global review of this topic. However, AA dwell on the regional aspects of this issue in South Africa (see lines 65-69 and lines 87-90) thus bringing it to be a mere regional survey. 

Line 79: Please replace “heavy metals” with “toxic metals”.

Materials and Methods

The organization of the M&M section is very confused and needs a rearrangement of the subsections.

Line 95: Why AA decided to start the bibliography consultation from 2011?

Line 98: Most of the study conducted in 2015 replaced the term heavy metals/hazardous metals with “toxic metals” or “trace elements”. Therefore, AA should revised the keyword used.

Results and Discussion

The exposition of the results give a lot of superfluous information (see lines 216-259) and does not contribute to the novelty of the study.

Line 135: AA did not take into account the analytical method used for every study considered in order to have a reliable comparison.

Lines 161-166 and lines 188-190: These parts are suitable for an introduction.

Lines 173-174: This is according to? AA refer only to HACCPs regulations. What about the Target Hazard Quotient, Provisional tolerable weekly intake and other risk assessment parameters recognized by FAO and other institutions?

Lines 180-186: This is speculative. AA refer only to wild angulates therefore, they should take into consideration the differences in bioaccumulation between birds and other wild animals.

AA should report data on numbers of game animals killed globally and the total number of hunters and the number of wild fauna killed per year. Furthermore, several studies stated that the extent of lead fragmentation depends on the type of bullet, its terminal velocity and the tissue penetrated  (Trinogga et al. 2019). Non-lead rifle bullets are designed not to fragment, thus avoiding contamination of the carcass. AA should give consideration on this in the discussion.

Conclusions

Lines 295-298: This is a summary of the discussion section. Please remove this sentence.

Author Response

Response to Reviewer 3

Comment 1: The review gives interesting outcomes on the global bibliography referred to the presence of toxic metals in game meat, encouraging the need to establish regulations relating to the presence of these contaminants. The novelty of the paper relies on the first systemic review on the incidence of toxic metals in this type of products in South Africa.

However, the present work presents several weaknesses regarding the complementarity of the scope of the study with what was reported in the main document. Furthermore, several improvements regarding the method of investigation used and the discussion of the results are needed.

Response 1: Thank you, for the positive comments we appreciate these. We also thank-you for your insightful suggestions/comments and believe that these will help in us submitting a quality review.

Comment 2: Introduction, the overall introduction is poor of references and gives unnecessary information, considering the aim of the work.

Response 2: Introduction: Noted, where possible, we have tried to improve the document and included more reference on the study subject and the discussions presented thereof. We have also attempted to align the Introduction more closely with the Aims of the review.

Comment 3 Lines 65-70: Considering the title of the MS, it could be a global review of this topic. However, AA dwell on the regional aspects of this issue in South Africa (see lines 65-69 and lines 87-90) thus bringing it to be a mere regional survey.

Response 3 thank you, while the monitoring and application of toxic metals in game meat or lack thereof is noted especially in developing countries including South Africa (Game intensive producing countries), it is important to focus the review globally as the situation is not totally different in most countries. The reference to South Africa and the region has been removed across the manuscript in an attempt to make it more relevant to a global readership. 

Comment 4 Line 79: Please replace “heavy metals” with “toxic metals”.

Response 4, heavy metals have been replaced with toxic metals across the manuscript.

Comment 5: Materials and Methods, the organization of the M&M section is very confused and needs a rearrangement of the subsections.

Response 5, thank you for the comment. The M&M section follows a standard way of literature identification. Where regulations / standards are sourced from organisations websites, these have been identified as additional records (n=7) and are also presented in Table 1. These records were combined with records identified through data base search where 242 were identified after all duplicates where removed. These formed the basis of the records screened of which 151 were removed as they fell outside the study objective. This left 91 records of which 49 records were further removed as they were not related to game and game meat production, leaving a total of 42 articles left that were included in the qualitative synthesis of records in the study subject within the study period. These 42 records are presented in table 1 and their objectives and finding are further discussed under discussions. We have tried to rephrase the M&M section to make it more clear on how the records were derived and the numbers selected.

Comment 6 Line 95: Why AA decided to start the bibliography consultation from 2011?

Response 6, thank you, indeed the search period is refined to work published between 2011 and 2021. This is done in the interest of size and current work. For the purpose of publishing this work, it was important that we report current work that has been published in recent years. Though work done pre 2011 has been read and some critical papers are included in the introduction and discussion sections. The systemic way of searching published articles as described under M&M proved to be vigorous and was able to provide refined articles for the search period. We are of the opinion that the information is current and relevant to current practices globally.

Comment 7 Line 98: Most of the study conducted in 2015 replaced the term heavy metals/hazardous metals with “toxic metals” or “trace elements”. Therefore, AA should revise the keyword used.

Response 7, thank you, the term heavy metals has been replaced with toxic metals across the manuscript, including in the keywords.

Comment 8 Results and Discussion

The expositions of the results give a lot of superfluous information (see lines 216-259) and do not contribute to the novelty of the study.

Response 8, thank you the information from the discussion of results has been refined and simplified (superfluous information as identified by yourself and other reviewers were removed) in line with the study objective.

Comment 9 Line 135: AA did not take into account the analytical method used for every study considered in order to have a reliable comparison.

Response 9, thank you, we appreciate the comment. The aim of the review is generally not to compare or draw a comparison on analytical methods used in literatures but to rather indicate the presence/ absence of toxic metals in game meat animals destined for consumption or the possible occurrences of these metals to an extent that it could cause harm to meat animals, that could have been contributed by the different killing methods used.

Comment 10 Lines 161-166 and lines 188-190: These parts are suitable for an introduction.

Response 10, thank you; sentences have been moved in the introduction of the document.

Comment 11 Lines 173-174: This is according to? AA refers only to HACCPs regulations. What about the Target Hazard Quotient, Provisional tolerable weekly intake and other risk assessment parameters recognized by FAO and other institutions?

Response 11, thank you, references to other toxic metals limiting guidelines is included in the manuscript. Table 1 has Codex limits of toxic metals. These limits are on meat products destined for human consumption. The objective of including this information was to estimating the available limits or controls of these toxic metals in meat products ready to be distributed for consumption. Indeed the weekly tolerable intake can be used to estimate weekly exposures per body weight. However, our view is these limits takes in to consideration different foods, different route of exposure that in general could give a weekly exposure data as combined.

Comment 12 Lines 180-186: This is speculative. AA refer only to wild angulates therefore; they should take into consideration the differences in bioaccumulation between birds and other wild animals.

Response 12, thank you, references of articles highlighting the risks of bioaccumulation and usage of birds of prey as indicators for environmental pollution has been added and discussed.

Comment 13 AA should report data on numbers of game animals killed globally and the total number of hunters and the number of wild fauna killed per year. Furthermore, several studies stated that the extent of lead fragmentation depends on the type of bullet, its terminal velocity and the tissue penetrated (Trinogga et al. 2019). Non-lead rifle bullets are designed not to fragment, thus avoiding contamination of the carcass. AA should give consideration on this in the discussion.

Response 13, thank you for the guidance. While we note that data on killing of game meat animals could be beneficial, its availability is speculative as a significant number of animals killed are also coming from informal trading (bush meat trade). Data on game meat animals killed has been included in the introduction to support the point that the consumption of game meat is increasing worldwide.

More references to new studies have been added and referred, while the study by Trinoga et al (2019), was cited, although the increased risk of fragmenting was discussed, we have added more work that has been conducted to highlight this threat.

We also take note of the new trend in having bullets that are non-lead and designed for specific functions, this is particularly relevant where marksmen (harvesters) use specially designed bullets to pierce the skull and then shatter destroying the brain, but not exiting to potentially wound other animals.

Comment Point 14 Conclusions

Lines 295-298: This is a summary of the discussion section. Please remove this sentence.

Response 14, thank you, the sentence has been removed